# Mechanistic Insights into the Adenosine A1 Receptor’s Positive Allosteric Modulation for Non-Opioid Analgesics

**DOI:** 10.3390/cells13242121

**Published:** 2024-12-21

**Authors:** Tal Weizmann, Abigail Pearce, Peter Griffin, Achille Schild, Maren Flaßhoff, Philipp Grossenbacher, Martin Lochner, Christopher A. Reynolds, Graham Ladds, Giuseppe Deganutti

**Affiliations:** 1Centre for Health and Life Sciences, Coventry University, Coventry CV1 5FB, UK; 2Department of Pharmacology, University of Cambridge, Tennis Court Road, Cambridge CB2 1PD, UK; 3Institute of Biochemistry and Molecular Medicine, University of Bern, 3012 Bern, Switzerland

**Keywords:** adenosine A1 receptor, BnOCPA, MIPS521, non-opioid analgesia, GPCRs

## Abstract

The adenosine A1 receptor (A_1_R) is a promising target for pain treatment. However, the development of therapeutic agonists is hampered by adverse effects, mainly including sedation, bradycardia, hypotension, or respiratory depression. Recently discovered molecules able to overcome this impediment are the positive allosteric modulator MIPS521 and the A1R-selective agonist BnOCPA, which are both potent and powerful analgesics with fewer side effects. While BnOCPA directly activates the A_1_R from the canonical orthosteric site, MIPS521 binds to an allosteric site, acting in concert with orthosteric adenosine and tuning its pharmacology. Given their overlapping profile in pain models but distinct mechanisms of action, we combined pharmacology and microsecond molecular dynamics simulations to address MIPS521 and BnOCPA activity and their reciprocal influence when bound to the A1R. We show that MIPS521 changes adenosine and BnOCPA G protein selectivity in opposite ways and propose a structural model where TM7 dynamics are differently affected and involved in the G protein preferences of adenosine and BnOCPA.

## 1. Introduction

The adenosine A1 receptor (A_1_R) is highly expressed in the brain cortex, cerebellum, and dorsal horn of the spinal cord [1], where adenosine mediates pain perception by modulating opioid neurotransmitter release [2,3]. It is, therefore, considered an emerging non-opioid target for novel painkillers [4]. However, the A_1_R is also present in the autonomic or somatic peripheral nervous systems, where its activation produces sedation, bradycardia, hypotension, and cardiorespiratory depression [5,6,7]. For this reason, it was considered undruggable until the recent discovery of the A_1_R agonist benzyloxy-cyclopentyladenosine (BnOCPA, Figure 1a and Appendix A) [8] and the A_1_R allosteric modulator [2-amino-4-(3,5-bis(trifluoromethyl)phenyl)thiophen-3-yl)(4-chlorophenyl)methanone] MIPS521 [9] (Figure 1a).

Allosteric modulation is the regulation of a protein’s activity through the binding of an effector molecule at a specific site on the protein, known as the allosteric site, which is distinct from the orthosteric (main) one [10]. This ternary interaction alters the protein’s conformation, thereby modulating its function. Allosteric modulators can either enhance (positive allosteric modulators—PAMs) or inhibit (negative allosteric modulators—NAMs) the activity of the protein. BnOCPA and MIPS521 have divergent action mechanisms but overlapping analgesic effects in vivo. BnOCPA’s direct action is orthosterically exerted through the preferential activation by the A_1_R of the subtype G_ob_ among the six G_αi/o_ proteins; the PAM MIPS521 enhances the effect of the endogenous agonist adenosine from a membrane-exposed allosteric binding site (Figure 1b).

Like the other class A G protein-coupled receptors (GPCRs), the A_1_R spans the cells’ plasma membrane seven times (transmembrane helices 1–7, TM1–7) and presents three intracellular loops (ICLs) and three extracellular loops (ECLs), which are critical for ligand binding [11,12,13]. The orthosteric site, located within the extracellular end of the transmembrane domain (TMD), stabilises the agonist binding mode thanks to five hydrogen bonds formed by N254^6.55^, E172^ECL2^, T277^7.42^, and H278^7.43^ (superscripts refer to the GPCR Ballesteros-Weinstein residue numbering [14]) as well as pi-pi stacking with F171^ECL2^ (Figure 1c). Agonist binding to the A_1_R stabilises the receptor in an active state, exposing the intracellular binding site for G_αi/o_ proteins. Several A_1_R conserved motifs are involved in the structural rearrangement upon activation [15], and many of them are located near the allosteric site for PAMs, which is external to the TMD and exposed to the membrane (Figure 1d). The hydrophobic MIPS521 binds to this accessory site through hydrophobic interactions with M283^7.48^, I19^1.42^, and V22^1.45^, and just one (buried) hydrogen bond with S246^6.47^. Contacts are also formed with G279^7.44^, which is the main player for TM7 flexibility [16].

BnOCPA’s signalling dissection has demonstrated that its ability to inhibit excitatory synaptic transmission without causing neuronal membrane hyperpolarisation is due to its propensity to couple to G_ob_ [8]. G_ob_ signalling activated by BnOCPA was validated as paramount for avoiding the depression of heart rate, blood pressure, or respiration in urethane-anaesthetised, spontaneously breathing adult rats. However, whether MIPS521 produces analgesia by switching adenosine selectivity towards G_ob_ intracellular pathways is not known. It is also unknown whether MIPS521 can further enhance BnOCPA’s unique signalling profile. In this work, we examined how these remarkable A_1_R ligands, BnOCPA and MIPS521, influence each other and what the structural bases are for their reciprocal interaction. We show that the dynamic of the A_1_R in complexes with BnOCPA or adenosine is differently affected by MIPS521 and suggest that TM7 is particularly important for transmitting BnOCPA’s chemical signal.

## 2. Materials and Methods

### 2.1. Chemical Synthesis

#### BnOCPA and MIPS521 Synthesis

The A_1_R agonist BnOCPA was synthesised according to our previously published procedure [17]. MIPS521 was synthesised according to the literature [18]. Both isolated compounds were unambiguously confirmed by nuclear magnetic resonance (NMR) and mass spectroscopy (MS) analyses and had a chemical purity of >95% according to high-performance liquid chromatography (HPLC) analysis.

### 2.2. Pharmacology

#### 2.2.1. cAMP Accumulation Assays

The measurement of ligand-induced cyclic adenosine monophosphate (cAMP) accumulation was performed in CHO-K1 cells stably expressing the A_1_R. Cells were cultured in F12 media supplemented with 10% foetal bovine serum (FBS), 1% antibiotic-antimycotic (AA) solution, and 600 μg/mL G418. Cells were maintained at 37 °C in 5% CO_2_. cAMP was measured using the LANCE*ultra* cAMP detection kit (Revvity) as per the manufacturer’s guidance and as previously described [8]. Cells were stimulated with adenosine, BnOCPA, cyclopentyladenosine (CPA), or 5-(N-ethylcarboxamido)adenosine (NECA) over the concentration range of 1 μM to 1 pM in the absence or presence of 10 μM–1 nM MIPS521. Cells were stimulated for 30 min, and plates were read using a Mithras LB 940 multimode microplate reader.

#### 2.2.2. TRUPATH G Protein Dissociation Assay

G_oα_ protein dissociation was measured using the TRUPATH biosensor platform [19], as described previously [8]. HEK293T cells, grown in Dulbecco’s modified Eagle medium (DMEM)/F12 Glutamax supplemented with 10% FBS and 1% AA solution, were transfected with 400 ng each of A_1_R, Gαo-Renilla Luciferase (Rluc)8, Gβ3, and Gγ8. After 24 h, cells were seeded at 50,000 cells per well of a white 96-well plate and assayed after another 24 h. Cells were stimulated with adenosine and BnOCPA over the concentration range of 10 μM to 10 pM in the absence or presence of 10 nM MIPS521. Plates were read for 15 min using a BRET2 module of a PheraStar at 60-s intervals. The BRET2 ratio after 10 min was used to calculate potency parameters.

#### 2.2.3. Data Analysis

Data were fitted using GraphPad Prism 10, the 3-parameter logistic equation, and the operational model of allosterism [20].

### 2.3. Computational Molecular Modelling

#### 2.3.1. Force Field and Ligands’ Parameters

The CHARMM36 [21,22]/CGenFF 3.0.1 [23,24,25] force field combination was employed in this work. Initial adenosine and MIPS521 (Figure 1a) force field topology and parameter files were obtained from the ParamChem webserver [23]. Force field parameters for BnOCPA were reported previously [8].

#### 2.3.2. System Preparation for Molecular Dynamics (MD) Simulations

The active state of the A_1_R in complex with adenosine and MIPS521 was retrieved from the Protein Data Bank database [26], entry 7LD3 [9]. The modelling of the A_1_R’s missing intracellular loop 3 (ICL3) and G_αi2_ alpha-helical domain (AHD) was not attempted. In our previous work [8], four different orientations of BnOCPA benzyloxy moiety were sampled (Appendix A). In this work, we started the simulations with BnOCPA in mode C, where the benzyloxy moiety was oriented towards Y271^7.36^ and inserted it into the A_1_R (7LD3) by superimposition with the A_1_R from our previous work.

Six different systems were prepared (Table 1): (*i*) two quaternary complexes were formed by agonists (adenosine or BnOCPA), A_1_R, MIPS521, and G_i2_; (*ii*) the same as (i) but omitting G_i2_; and (*iii*) the same as (i) but omitting MIPS521.

Hydrogen atoms were added using pdb2pqr [27] and propka [28] software (considering a simulated pH of 7.0); the protonation of titratable side chains was checked by visual inspection. The resulting receptors were separately inserted in a 1-palmitoyl-2-oleyl-sn-glycerol-3-phosphocholine (POPC) bilayer (previously built using the VMD Membrane Builder plugin 1.1, Membrane Plugin, Version 1.1. at http://www.ks.uiuc.edu/Research/vmd/plugins/membrane/ accessed on 11 November 2023), through an insertion method [29]. The receptor orientation was obtained by superposing the coordinates on the corresponding structure retrieved from the OPM database [30]. Lipids overlapping the receptor transmembrane helical bundle were removed, and TIP3P water molecules [31] were added to the simulation box employing the VMD Solvate plugin 1.5 (Solvate Plugin, Version 1.5. at http://www.ks.uiuc.edu/Research/vmd/plugins/solvate/ accessed on 11 November 2023). Finally, overall charge neutrality was reached by adding Na^+^/Cl^−^ counter ions up to the final concentration of 0.150 M), using the VMD Autoionize plugin 1.3 (Autoionize Plugin, Version 1.3. at http://www.ks.uiuc.edu/Research/vmd/plugins/autoionize/ accessed on 11 November 2023).

#### 2.3.3. System Equilibration and Molecular Dynamics Settings

The MD engine ACEMD3 [32] was employed for both the equilibration and productive simulations. The equilibration was achieved in isothermal–isobaric conditions (NPT) using the Berendsen barostat [33] (target pressure 1 atm) and the Langevin thermostat [34] (target temperature 300 K) with low damping at 1 ps^−1^ and an integration time step of 2 fs. An initial 1 kcal mol^−1^ Å^−2^ restraint was applied to protein, ligands, and lipid phosphorus atoms. Restraints not involving the ligands were gradually released at 4 ns (phosphorus atoms), 60 ns (protein atoms other than Cα atoms), and 80 ns (protein Cαatoms). A further 20 ns (in the absence of the G_i2_ protein) or 40 ns (in the presence of the G_i2_ protein) of equilibration were run with restraints applied only on ligand atoms. Productive trajectories (Table 1) were computed with an integration time step of 4 fs in the canonical ensemble (NVT). The target temperature was set at 300 K, using a thermostat damping of 0.1 ps^−1^; the M-SHAKE algorithm [35,36] was employed to constrain the bond lengths involving hydrogen atoms. The cut-off distance for electrostatic interactions was set at 9 Å, with a switching function applied beyond 7.5 Å. Long-range Coulomb interactions were handled using the particle mesh Ewald summation method (PME) [37] by setting the mesh spacing to 1.0 Å.

#### 2.3.4. Molecular Dynamics Trajectories: Analysis

Root mean square deviations (RMSDs) and root mean square fluctuations (RMSFs) were computed using VMD [38] and dihedral angles with MDAnalysis [39]. Interatomic contacts were detected using the GetContacts scripts tool (https://getcontacts.github.io accessed on 4 June 2022). Contacts and hydrogen bond persistency were quantified as the percentage of frames (over all the frames obtained by merging the different replicas) in which protein residues formed contacts or hydrogen bonds with the ligand.

For the network analysis, the three replicas simulated for each complex were merged (3 μs for each system). Network and community analyses [40] (code available at http://faculty.scs.illinois.edu/schulten/software/networkTools/index.html accessed on 21 March 2024) were performed within the VMD environment, considering the Cα atoms of A_1_R as nodes. A network was formed by an ensemble of nodes interconnected by edges. Edges connected pairs of (non-consecutive) nodes if the corresponding residues were in contact (within 4.5 Å) for at least 75% of the frames. The resulting dynamical network was weighted by considering the probability w_ij_ of information transfer across the edge connecting two nodes *I* and *j*, which were calculated using the following equation:W_ij_ = −log(|C_ij_|)
where |*C_i_*_j_ | measures the correlation values of motion between the two residues during the simulation (Appendix A).

### 2.4. G Protein-Coupled Receptor Numbering System

Throughout the manuscript, the Ballesteros-Weinstein residue numbering system for GPCRs [14] was adopted.

## 3. Results

### 3.1. MIPS521 Similarly Enhances the Potency of Different A_1_R Agonists

In a cAMP inhibition assay, MIPS521 acts as an agonist of the A_1_R, consistent with previous observations [9]. We sought to examine its effects against different A_1_R orthosteric agonists: the non-selective adenosine and N-ethylcarboxamindoadenosine (NECA), and the A_1_R selective agonists N^6^-cyclopentyladenosine (CPA) and BnOCPA. MIPS521, tested across the concentration range of 10 nM to 10 μM, potentiated all four orthosteric agonists to similar extents (Figure 1e and Appendix A, Table 2). The cooperativity operator Logαβ [41], incorporating allosteric effects on affinity (α) and efficacy (β), computed for the four agonists resulted in values between 1.17 and 1.57 (Figure 1f, Table 2), confirming a comparable effect of MIPS512 on all the agonists tested.

### 3.2. MIPS521 Requires G_i/o_ for A_1_R Stable Binding

To retrieve structural insights into the allosteric mechanism putatively triggered by MIPS521 on adenosine and BnOCPA, six different systems were subjected to MD simulations (Table 1): (*i*) two quaternary complexes formed by an agonist (adenosine or BnOCPA), A_1_R, MIPS521 and G_i2_; (*ii*) the same as in (i) but omitting MIPS521; or (*iii*) the as in (i) but omitting G_i2_.

During MD simulations of MIPS521 in a ternary complex with the A_1_R and adenosine (Appendix A), MIPS521 remained bound to the A_1_R in two out of three replicas and dissociated from the receptor in Replica 3 after 550 ns. This latter MIPS521′s unstable behaviour was observed in a ternary complex with the A_1_R and BnOCPA (Appendix A), where MIPS521 either transiently (Replicas 1 and 2) or permanently (Replica 3) unbound from the A_1_R. On the other hand, in a quaternary complex with the A_1_R, G_i2_, and either adenosine or BnOCPA (Appendix A), MIPS521 remained stably bound to the A_1_R (RMSD 2.32 ± 0.87 Å and 2.50 ± 1.72 Å, respectively) except for transient conformational rearrangements highlighted by RMSD values of 6–8 Å. These computational results suggest that MIPS521 binding is allosterically favoured by G_i/o_, thanks to the stabilisation provided by TM6 in the active conformation, which is necessary for shaping the MIPS521 membrane-facing binding site (Figure 1d).

### 3.3. BnOCPA Binding Mode Is Affected by G_i2_ and MIPS512 Through TM7 Dynamics

Focusing on the simulated agonists, adenosine experienced similar fluctuations within the A_1_R orthosteric site in a ternary complex with the A_1_R and MIPS521 (Appendix A) or with the A_1_R and G_i2_ (Appendix A) (RMSD 1.46 ± 0.49 Å and 1.43 ± 0.61 Å, respectively). However, it was stabilised in the quaternary complex with A_1_R, MIPS521 and G_i2_ (1.03 ± 0.41 Å, Appendix A). BnOCPA in a ternary complex with the A_1_R and MIPS521 (Appendix A) remained stably bound in the initial conformation during two replicas out of three, with Replica 3 characterised by a rotation of the benzyloxy-cyclopentyl group from mode C to modes A or B (Appendix A) after 0.5 μs. Interestingly, BnOCPA’s conformational rearrangements from mode C to modes A and B were sampled in all three replicas performed on the ternary complex comprising A_1_R and G_i2_ (i.e., without MIPS512) within 0.8 μs (Appendix A) but not when MIPS521 was present in the quaternary complex (Appendix A). In this latter stable configuration, the RMSD of BnOCPA remained stably low (1.54 ± 0.53 Å), and the agonist benzyloxy-cyclopentyl group was stably oriented in mode C (Appendix A). This confirms that the allosteric effect of MIPS521 on adenosine likely depends on the presence of G_i2_ and whether G_i2_ alone or with MIPS521 differently influences BnOCPA’s orthosteric dynamics.

To investigate the structural bases involved in the different allosteric effects exerted by G_i2_ alone and in the co-presence of MIPS521, we analysed the flexibility of the A_1_R Cα during the MD simulations in complex with adenosine or BnOCPA in these two divergent situations (Figure 2a–d). Regardless of the agonist bound, a comparison between A_1_R backbone flexibility with and without MIPS521 showed the stabilisation of the intracellular end of TM6 and the extracellular loop 2 (ECL2) due to PAM binding. This is consistent with the MIPS521 binding site partially involving TM6 and, therefore, a reduction in TM6 and the fluctuation favoured by the hydrogen bond with S246^6.47^. An ECL2 stabilisation triggered by MIPS521 was also observed by Solano et al. by employing enhanced sampling MD simulations [16]. A difference, evident only in the BnOCPA-bound A_1_R without MIPS521, is represented by the higher flexibility in TM7 (residues G279^7.44^ and N280^7.45^) and TM2 (e.g., N70^2.65^) (Figure 2c, Appendix A). The binding of MIPS521 to the A_1_R in complex with BnOCPA and G_i2_ (Figure 2d) had the effect of reducing the fluctuations of G279^7.44^ and N280^7.45^. This was likely due to direct interactions with TM7 and TM2, caused by an allosteric effect that could involve the stabilisation of BnOCPA in binding mode C (Appendix A).

The side chain conformational analysis of the residues connecting MIPS521 and the agonist binding site revealed a divergent rotameric distribution of T277^7.42^ and H278^7.43^, both pivotal for A_1_R activation [42], whether MIPS521 was co-present with G_i2_ or absent (Appendix A). The conformational shift of T277^7.42^ and H278^7.43^ upon MIPS521 binding was paralleled by increased interactions with BnOCPA, which was particularly evident for T277^7.42^ (Figure 2e). No significant differences in side chain conformation or interactions were observed for adenosine (Figure 2e and Appendix A).

Taken together, these results suggest a structural communication between the MIPS521 and BnOCPA binding sites that involve TM7, not parallel to the endogenous agonist adenosine.

### 3.4. G_i2_ and MIPS512 Favour Different Interactions Within the BnOCPA-Bound A_1_R

Given the role of TM7 suggested by the previous analysis, we focused on A_1_R intramolecular interactions involving TMD residues at the level of the MIPS521 binding site, particularly near the conserved residues D55^2.50^ and N284^7.49^, which are involved in receptor activation [42]. When the adenosine was bound to A_1_R, either in a ternary complex with G_i2_ or a quaternary complex with MIPS521 as well, no significant changes in contacts were evident for N280^7.45^, S281^7.46^, or N284^7.49^ (Figure 3a). Significant increases in interactions between TM7 and TM2 or TM1 (i.e., N280^7.45^-W247^6.48^, S281^7.46^-D55^2.50^, S281^6.46^-N27^1.50^, and N280^7.45^-N284^7.49^) were observed with BnOCPA bound to A_1_R:G_i2_:MIPS521 compared to A_1_R:G_i2_ (Figure 3a), paralleled by a reduction in intrahelical TM7 contacts (i.e., L276^7.41^-N280^7.45^ and S281^7.46^-N284^7.49^) suggesting the strengthening of the intramolecular interactions between TM7 and TM2 or TM1 due to the PAM.

A network analysis was performed to pinpoint the A_1_R residues involved in the flow of structural information through the receptor’s TM7 under the different simulated conditions (Figure 3b–e). Interestingly, the presence of MIPS521 bound to A_1_R, in turn, and in complex with adenosine, concentrated the structural information in the middle of TM7 and involved F275^7.40^, H278^7.43^, and S281^7.46^, in addition to G279^7.44^ (Figure 3c), compared to the sparse configuration in the absence of MIPS521 (Figure 3b). Contrarily, the TM7 of BnOCPA-bound A_1_R was deactivated by MIPS521, as shown by the marked reduction in structural information through TM7 (Figure 3d,e). In the latter case, BnOCPA was able to trigger a dense and extended network of correlated movement involving the TM7 of the A_1_R bound to G_i2_ in the absence of MIPS521 (Figure 3d), which instead turned off the structural communication in the extracellular half of the helix, de facto isolating the orthosteric binding site from the intracellular side of TM7 (Figure 3e).

To assess any difference in the G protein subtype activation by MIPS521, possibly due to the intramolecular interaction network stabilised by MIPS521 and BnOCPA, we performed TRUPATH experiments detecting G_βγ_ and its dissociation from G_oaα_ or G_obα_. A significant enhancement of the G_ob_ activation by adenosine (Figure 3f and Appendix A, Table 3) and G_oa_ activation by BnOCPA (Figure 3g and Appendix A, Table 3**)** was detected at 10 nM of MIPS521. Consistent with the different interactions in the network pinpointed by MD simulations, these results show a divergent effect of MIPS521 on the G_i/o_ protein subtype’s selectivity of two agonists. BnOCPA displayed a loss of G_ob_ selectivity due to MIPS512, whereas adenosine gained G_ob_ selectivity.

## 4. Discussion

In the present work, we first showed that adenosine and its derivatives, CPA, BnOCPA, and NECA, are consistently enhanced by MIPS521, as demonstrated by the increase in pEC_50_ and the similar value of the allosteric operator Logαβ. Microsecond MD simulations of the A_1_R in complex with adenosine or BnOCPA, with or without G_i2_ or MIPS521, indicated that the PAM requires an active state of TM6 conformation stabilised by G_i2_ to bind to the receptor’s external binding site. These results, in agreement with previously reported Gaussian-accelerated MD simulations by Miao and co-workers [9], also showed that MIPS521 binding reciprocally affects TM6 flexibility when A_1_R engages G_i2_, likely stabilising the G protein [9]. Given that this reciprocal allosteric communication between the intracellular effector and the PAM is observed for both adenosine and BnOCPA, we hypothesise that the TM6 stabilisation can play a role in the generic, unselective activation of any Gi/o protein, which would explain the common increase in cAMP inhibition for both adenosine and BnOCPA.

In the absence of MIPS521, the BnOCPA:A_1_R:G_i2_ complex displayed high TM7 flexibility that was not paralleled by adenosine in the same conditions. The high intracellular plasticity of TM7 was also evident in our previous computational characterisation of BnOCPA [8] in the binary complex with A_1_R (i.e., without Gi2 and in the absence of MIPS521). The hypothesis that TM7 is pivotal for the unique pharmacological profile of BnOCPA on that occasion [8] was partially supported by the mutagenesis of the A_1_R residues located at the TM7/H8 interface level. Indeed, the mutation of R291^7.56^, I292^8.47^, Q293^8.48^, and K294^8.49^ to alanine differentially affected agonist efficacy against stimulated cAMP production. Our new MD-based results strengthen the hypothesise that a direct effect of MIPS521 can be had on TM7 at the level of the A_1_R residues G279^7.44^ and N280^7.45^ that are transmitted to the orthosteric binding site through TM7, with the impact of altering T277^7.42^ and H278^7.43^ side chain dynamics and stabilising the BnOCPA binding mode in the C orientation. The divergent structural response of TM7 to MIPS521 when adenosine or BnOCPA were bound to the receptor might underlie the opposite G_oa_ and G_ob_ selectivity enhancement produced by the PAM. This can be framed in the probe-dependency of allosteric modulation, where the response of a receptor to an allosteric modulator varies based on the nature of the orthosteric ligand. In GPCRs, probe-dependency is often observed because the orthosteric ligands stabilise different receptor states, affecting how PAMs influence receptor activation and signal transduction [43]. MIPS521, reducing the efficacy of BnOCPA by activating G_ob_, would not represent an obstacle to developing A_1_R agonist painkillers because they can independently stimulate the G_ob_ pathway (Figure 4), as demonstrated by BnOCPA. On the other hand, MIPS (and related PAMs) depend on endogenous agonist adenosine rather than xenobiotic compounds for producing in vivo analgesia.

Importantly, the α subunit, pivotal for the trimeric G protein coupling [44] of G_oa_ and G_ob_, differs in just 19 residues, none of which are part of the main interface engaging the A_1_R. We propose that the selectivity towards one or the other of these two G_i/o_ isoforms is due to a concerted mechanism involving the receptor subdomains TM6 and TM7.

## 5. Conclusions

Understanding the structural bases for agonist-specific G_i/o_ activation in a dynamic molecular context could speed up the long-awaited rational development of novel non-opioid analgesics based on purinergic signalling. Here, we propose new molecular details of the A_1_R activation by BnOCPA and its modulation by MIPS521, which are both considered promising lead compounds. We previously demonstrated that the analgesic effect of BnOCPA is due to G_ob_ selectivity. Here, we report that MIPS512 in vivo analgesia can be due to the sensitisation of adenosine towards G_ob._ However, the reciprocal structural communication through TM7, as suggested by our computational modelling, between BnOCPA and MIPS521 could reduce BnOCPA potency towards G_ob_ and, in turn, its efficacy as an analgesic. Our key finding that the A_1_R-dependent G_ob_ signalling path activation, which is necessary to mitigate on-target side effects, can be triggered either directly by agonists or allosterically by PAMs broadens the approaches available for rationally designing new and safer non-opioid drugs.

## Figures and Tables

**Figure 1 cells-13-02121-f001:**
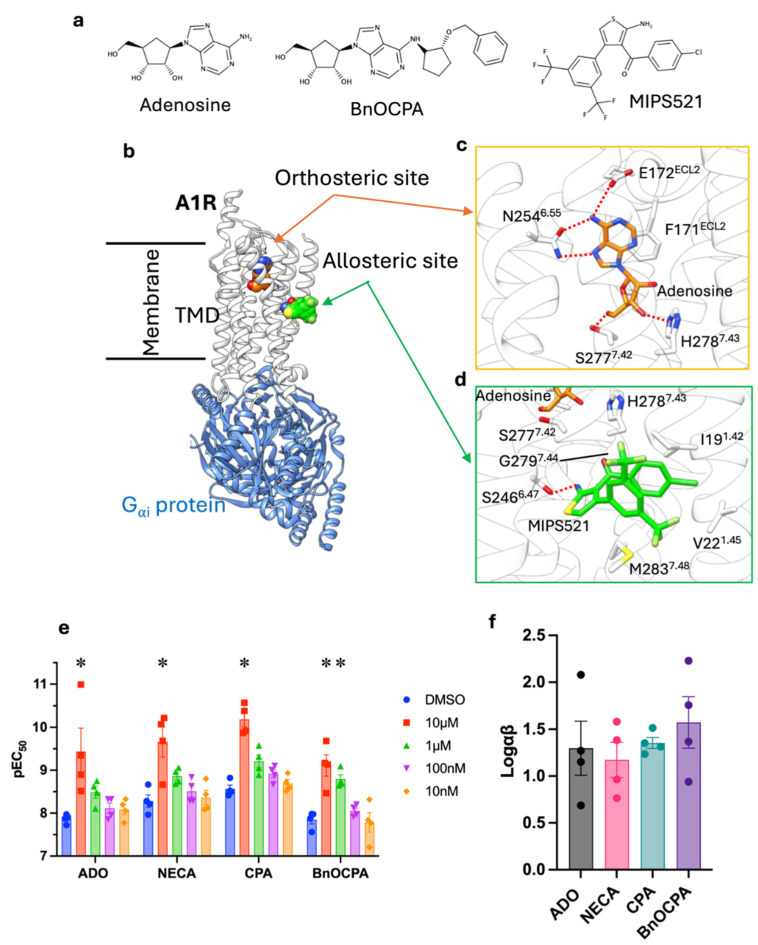
(**a**) Structure of the A_1_R ligand simulated in this study: the agonists adenosine and BnOCPA and the positive allosteric modulator MIPS521; (**b**) membrane view of the orthosteric and allosteric binding site of the A_1_R (white ribbon) bound to the G_αi_ protein (blue ribbon); (**c**,**d**) detailed binding mode of adenosine (orange) and MIPS521 (green) within their respective binding sites; (**e**) cAMP assay pEC50 of the A_1_R agonists CPA, BnOCPA, adenosine (ADO) and NECA at increasing concentrations of MIPS521 (10 nM, orange; 100 nM, purple; 1 μM, green, *: *p* = 0.0042 for BnOCPA; 10 μM, red, *: *p*  <  0.0001 for all the agonists tested). (**f**) MIPS allosteric operator Logαβ measured for CPA (cyan), BnOCPA (purple), ADO (grey), and NECA (pink).

**Figure 2 cells-13-02121-f002:**
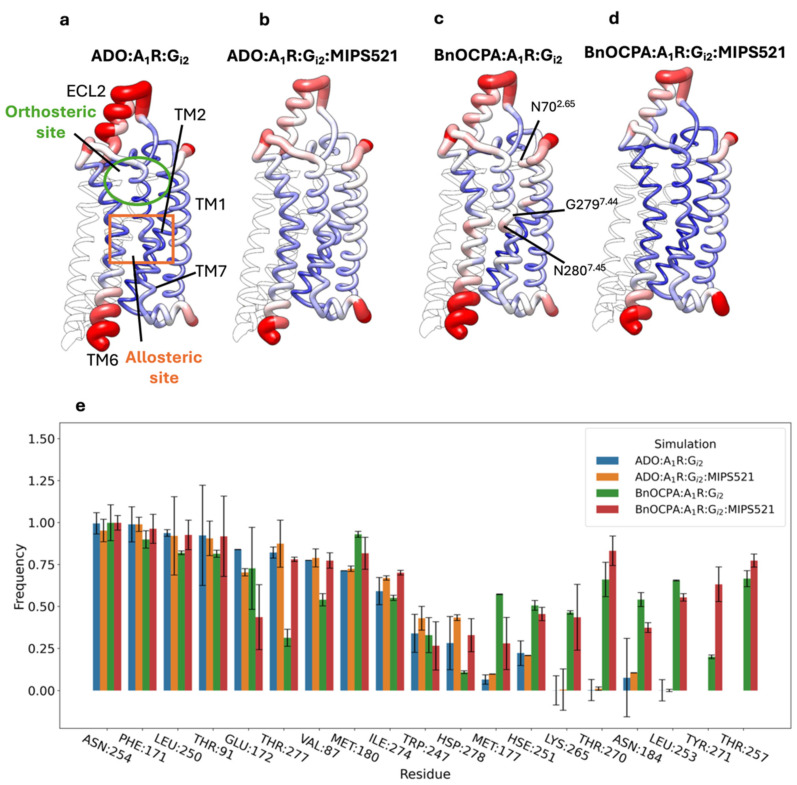
(**a**–**d**) A_1_R Cα root mean square fluctuations (RMSFs) plotted on the receptor backbone, colour-coded, and with ribbon thickness proportional to the RMSF value for (**a**) ADO:A_1_R:G_i2_, (**b**) ADO:A_1_R:G_i2_:MIPS521, (**c**) BnOCPA:A_1_R:G_i2_, (**d**) BnOCPA:A_1_R:G_i2_:MIPS521; the orange rectangle in (**a**) indicates the MIPS521 binding site position. (**e**) Contacts between the A_1_R and adenosine or BnOCPA in a ternary or quaternary complex with G_i2_ and MIPS521.

**Figure 3 cells-13-02121-f003:**
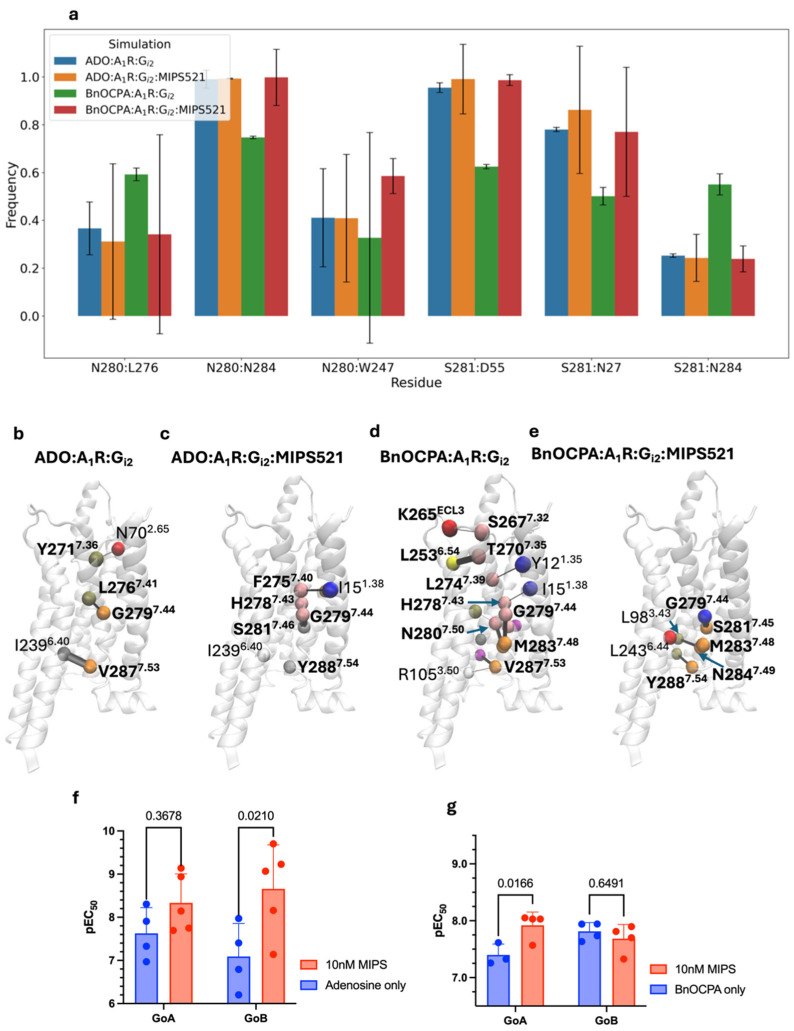
(**a**) Intramolecular contacts between the A_1_R residues on TM7 and TM1 or TM2 during MD simulations of the A_1_R in a ternary complex with G_i2_ and adenosine or BnOCPA or in a ternary or quaternary complex with G_i2_ and MIPS521; standard deviations from 3 replicas are reported. (**b**–**e**) Network analysis of TM7 during MD simulations of the A_1_R in ternary complex with G_i2_ and adenosine or BnOCPA, or a ternary or quaternary complex with G_i2_ and MIPS521; Cα carbons are shown as van der Waals spheres (nodes) and node edges as black lines of thickness proportional to the structural information computed. (**f**,**g**) G_oa_ and G_ob_ activation of TRUPATH pEC50 for adenosine and BnOCPA**.**

**Figure 4 cells-13-02121-f004:**
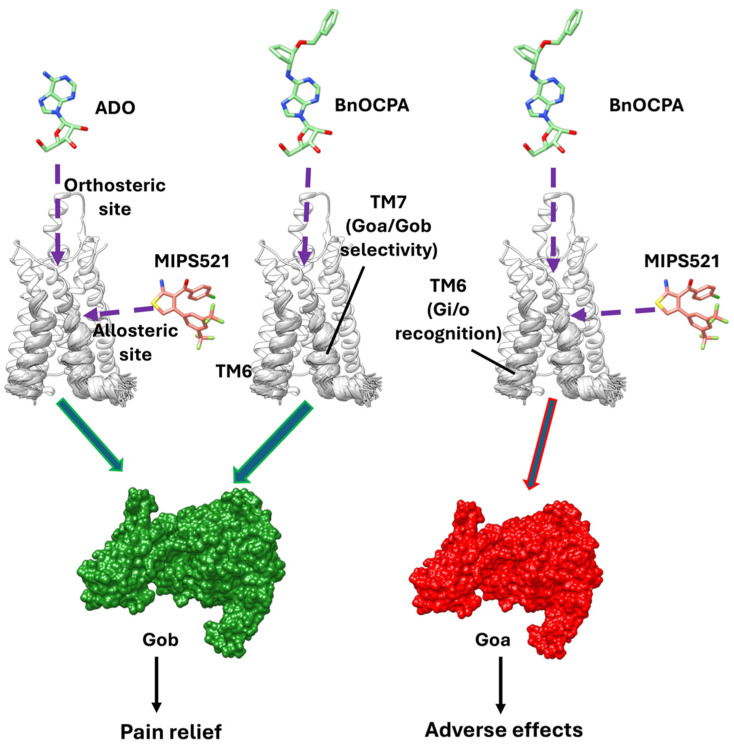
MIPS521 probe-dependent effect and proposed model of G_ob_ selectivity mediated by TM7. Adenosine (ADO) and MIPS521 favour the A_1_R selectivity towards G_ob_, in analogy with BnOCPA alone. MIPS521 shifts A_1_R signalling towards G_oa_, potentially producing undesired effects. TM7 dynamics, differently affected by adenosine and BnOCPA, are proposed to be crucial for MIPS521 probe-dependent signalling and, hence, for G_oa_/G_ob_ selectivity.

**Table 1 cells-13-02121-t001:** Summary of the MD simulations performed.

Complex Components	Coordinates Source	# Replicas (Total MD Sampling)/μs	Initial System Dimensions/Å
A_1_R	PDB 7LD3	3 × 1 (3)	111 × 101 × 156 167,339 atoms
Adenosine	PDB 7LD3		
MIPS521	PDB 7LD3		
G_αi2_	PDB 7LD3		
A_1_R	PDB 7LD3	3 × 1 (3)	111 × 101 × 156 167,300 atoms
Adenosine	PDB 7LD3		
G_αi2_	PDB 7LD3		
A_1_R	PDB 7LD3	3 × 1 (3)	81 × 81 × 106 64,639 atoms
Adenosine	PDB 7LD3		
MIPS521	PDB 7LD3		
A_1_R	PDB 7LD3	3 × 1 (3)	111 × 101 × 156 167,500 atoms
BnOCPA	Mode C (Ref [8])		
MIPS521	PDB 7LD3		
G_αi2_	PDB 7LD3		
A_1_R	PDB 7LD3	3 × 1 (3)	111 × 101 × 156 167,461 atoms
BnOCPA	Mode C (Ref [8])		
G_αi2_	PDB 7LD3		
A1R	PDB 7LD3	3 × 1 (3)	81 × 81 × 106 64,797 atoms
BnOCPA	Mode C (Ref [8])		
MIPS521	PDB 7LD3		

**Table 2 cells-13-02121-t002:** pEC_50_ and Logαβ values show comparable allosteric modulation for all orthosteric agonists tested. Values calculated from cAMP accumulation assays are reported as the mean ± SEM of *n* = 4 replicates. Significance from DMSO was determined through a Two-Way ANOVA with Dunnett’s post hoc test for multiple comparisons.

Agonist	DMSO	10 μM	1 μM	100 nM	10 nM	Logαβ
Adenosine	7.85 ± 0.07	9.46 ± 0.55 *	8.51 ± 0.14	8.12 ± 0.12	8.09 ± 0.12	1.30 ± 0.29
NECA	8.31 ± 0.17	9.61 ± 0.32 *	8.69 ± 0.09	8.51 ± 0.14	8.36 ± 0.17	1.17 ± 0.19
CPA	8.56 ± 0.09	10.22 ± 0.14 *	9.22 ± 0.15	8.92 ± 0.09	8.69 ± 0.09	1.35 ± 0.06
BnOCPA	7.76 ± 0.13	9.34 ± 0.36 *	8.82 ± 0.10 *	8.05 ± 0.07	7.79 ± 0.23	1.57 ± 0.28

* *p* < 0.05.

**Table 3 cells-13-02121-t003:** pEC50 values for Gαo protein dissociation in response to the stimulation with ADO or BnOCPA in the presence of either 0.1% DMSO or 10 nM MIPS521. Mean values are reported as the ± SEM of n individual experiments, where n is between 4 and 5. The significance of DMSO-treated cells was determined using a Two-Way ANOVA.

	G_oa_		G_ob_	
Agonist	DMSO	10nM MIPS521	DMSO	10nM MIPS521
Adenosine	7.63 ± 0.30	8.33 ± 0.30	7.09 ± 0.38	8.66 ± 0.46 *
BnOCPA	7.40 ± 0.11	7.92 ± 0.12 *	7.81 ± 0.08	7.65 ± 0.13

* *p* < 0.05.

## Data Availability

Data are available upon request.

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
