# Peer review of "Mechanistic Insights into the Adenosine A1 Receptor’s Positive Allosteric Modulation for Non-Opioid Analgesics"

_cells, 2024, doi:10.3390/cells13242121_

Round 1

Reviewer 1 Report

Comments and Suggestions for Authors

In the proposed article, the authors aimed to investigate the molecular mechanisms underlying the activity and interaction of the positive allosteric modulator MIPS521 and the A1R-selective agonist BnOCPA on the adenosine A1 receptor to develop safer, non-opioid analgesics.

The manuscript's abstract is dense with detailed information, which, while comprehensive, may overwhelm readers. Simplifying it by focusing on the core findings (such as the role of MIPS521 in enhancing A1 receptor function and its potential for non-opioid analgesia and their broader implications) would improve accessibility.

In the Introduction section, including brief lay explanations (e.g., describing allosteric modulation as a process where certain molecules enhance receptor activity by binding at a distinct site from the main active site) would make the article more approachable to a broader audience. Balancing technical accuracy with readability will enhance its impact and comprehension.

The manuscript provides substantial insights into the interplay between orthosteric agonists and the positive allosteric modulator MIPS521. However, some conclusions remain speculative, particularly regarding TM7’s role and G-protein selectivity. These should be explicitly acknowledged as limitations. For instance, the suggestion that MIPS521 alters G-protein subtype preferences in the A1 receptor (A1R), enhancing adenosine selectivity for Gob and reducing BnOCPA’s Gob potency, is primarily based on computational simulations and requires additional experimental validation.

TM7 is highlighted as a key structural player in the interaction between BnOCPA, MIPS521, and A1R. While intriguing, the evidence is largely inferred from molecular dynamics (MD) simulations. Including more detailed analyses from the existing MD data, such as interaction energy calculations between TM7 and BnOCPA/MIPS521, or conducting additional computational experiments to confirm TM7's flexibility changes and influence on G-protein coupling, would strengthen this claim. Experimental approaches, such as site-directed mutagenesis or cryo-EM studies, are also recommended to validate the role of TM7 residues identified in the simulations.

While the manuscript’s structural insights are well-presented, they could better connect to therapeutic applications. Simplifying the technical density in the results and discussion sections and emphasizing key findings through figures or tables would make the study more accessible to readers outside the immediate field of GPCR pharmacology.

Finally, the conclusions are comprehensive but could more effectively emphasize the practical implications of the findings and the broader impact on drug development. Highlighting the translational potential of targeting the A1R with MIPS521 and BnOCPA will make the study more impactful.

As an additional point, ensure uniformity in abbreviations (e.g., "BnOCPA" vs. "Bnocpa") and provide a complete abbreviation list at the end of the manuscript.

Considering all these observations, I believe the article could be considered for publication after major revision.

Author Response

In the proposed article, the authors aimed to investigate the molecular mechanisms underlying the activity and interaction of the positive allosteric modulator MIPS521 and the A1R-selective agonist BnOCPA on the adenosine A1 receptor to develop safer, non-opioid analgesics.

We thank the reviewer for the appraisal of our manuscript and the suggestions provided.

  1. The manuscript's abstract is dense with detailed information, which, while comprehensive, may overwhelm readers. Simplifying it by focusing on the core findings (such as the role of MIPS521 in enhancing A1 receptor function and its potential for non-opioid analgesia and their broader implications) would improve accessibility.

The abstract has been edited to simplify the jargon, improve the readability and provide more background.

  1. In the Introduction section, including brief lay explanations (e.g., describing allosteric modulation as a process where certain molecules enhance receptor activity by binding at a distinct site from the main active site) would make the article more approachable to a broader audience. Balancing technical accuracy with readability will enhance its impact and comprehension. Also, add the 2 refs suggested by the Editor.

The two suggested references and a piece of text introducing the definition of allostery (lines 42-47) have been added to the introduction.

  1. The manuscript provides substantial insights into the interplay between orthosteric agonists and the positive allosteric modulator MIPS521. However, some conclusions remain speculative, particularly regarding TM7’s role and G-protein selectivity. These should be explicitly acknowledged as limitations. For instance, the suggestion that MIPS521 alters G-protein subtype preferences in the A1 receptor (A1R), enhancing adenosine selectivity for Gob and reducing BnOCPA’s Gob potency, is primarily based on computational simulations and requires additional experimental validation.

In the Discussion (lines 346-350), the revised version cites some of our previous mutagenesis results that partially corroborate our claim. We have also stressed that some results are based on MD simulations so should be considered hypothetical.

  1. TM7 is highlighted as a key structural player in the interaction between BnOCPA, MIPS521, and A1R. While intriguing, the evidence is largely inferred from molecular dynamics (MD) simulations. Including more detailed analyses from the existing MD data, such as interaction energy calculations between TM7 and BnOCPA/MIPS521, or conducting additional computational experiments to confirm TM7's flexibility changes and influence on G-protein coupling, would strengthen this claim. Experimental approaches, such as site-directed mutagenesis or cryo-EM studies, are also recommended to validate the role of TM7 residues identified in the simulations

We have added a network analysis to pinpoint the key residues on TM7 responsible for allosteric communication between the extracellular and intracellular sides of A1R (Figure 3b-e). It clearly shows that MIPS521 can alter the structural information passing through TM7 in an agonist-dependent way, supporting the overall mechanistic model proposed.

We also agree with the Reviewer regarding the experimental validation and its possible role in strengthening our hypothesis. As per point 3, previous site-directed mutagenesis partially validates the role of TM7 in A1R G protein coupling. However, NMR studies might provide clearer insights into A1R dynamics, which we hope to address in future work. 

  1. While the manuscript’s structural insights are well-presented, they could better connect to therapeutic applications. Simplifying the technical density in the results and discussion sections and emphasizing key findings through figures or tables would make the study more accessible to readers outside the immediate field of GPCR pharmacology.

We thank the Reviewer for the comment. The Discussion has been amended to better frame our results in the context of allosterism and to comprehend a new figure (Figure 4) summarising the findings reported in the manuscript. We report relatively straightforward analyses such as RMSD, RMSF, interactions and dihedral angles. As such, we respectfully prefer not to modify the bulk of the Results.

  1. Finally, the conclusions are comprehensive but could more effectively emphasize the practical implications of the findings and the broader impact on drug development. Highlighting the translational potential of targeting the A1R with MIPS521 and BnOCPA will make the study more impactful.

Thanks for the suggestion. We have added a final phrase to the Conclusion to highlight the practical implications of our findings.

  1. As an additional point, ensure uniformity in abbreviations (e.g., "BnOCPA" vs. "Bnocpa") and provide a complete abbreviation list at the end of the manuscript.

We thank the reviewer for the careful reading. The abbreviations in the figures are now consistent with the text. Also, a list of abbreviations has been added.

Reviewer 2 Report

Comments and Suggestions for Authors

The authors investigated the mechanisms by which two compounds BnOCPA and MIPS521 modulate A1R to achieve analgesia without the side effects associated with A1R activation. Considering A1R’s involvement in various pain mechanisms and its lack of opioid-like addictive properties, this study is an important contribution to the field. While the manuscript is well-written and provides strong molecular insights, there are areas where additional data or discussion could enhance the findings.

  1. The study demonstrates molecular changes and receptor selectivity. However, predicting the absence of side effects based solely on these findings is challenging. Functional testing in relevant biological systems would be important to confirm whether these compounds avoid predicted side effects, such as bradycardia, hypotension, or sedation. The authors could consider testing these compounds in vascular smooth muscle cells or neuronal models to address this. If testing these models is not feasible, acknowledging this limitation in the discussion would be valuable.
  2. Again, functional experiments testing the outcomes of MIPS521 and BnOCPA in pain-related cellular models would significantly strengthen the manuscript. For example, assays measuring neuronal firing rates or calcium signaling in DRG neurons under pain-like conditions could demonstrate their analgesic potential: authors could test the effect of combination of MIPS521 and BnOCPA on relieving simulated pain states. The results would provide critical evidence to support their proposed therapeutic effects.
  3. The authors state that the interaction between MIPS521 and BnOCPA may reduce the efficacy of BnOCPA in activating Gob proteins. This trade off could be a limitation and the manuscript would benefit from further discussion on how this affects the therapeutic potential of these compounds.
  4. Figures 2e and 3a do not have dot plots or statistical significance annotations. Also, the number of replicates and experimental repetitions is not clearly indicated for all figures. 
  5. The methods section includes a substantial amount of technical details, but some descriptions are difficult to follow. Reorganizing or rephrasing parts of this section could improve clarity and accessibility for the reader.

Author Response

The authors investigated the mechanisms by which two compounds BnOCPA and MIPS521 modulate A1R to achieve analgesia without the side effects associated with A1R activation. Considering A1R’s involvement in various pain mechanisms and its lack of opioid-like addictive properties, this study is an important contribution to the field. While the manuscript is well-written and provides strong molecular insights, there are areas where additional data or discussion could enhance the findings.

We are grateful to the reviewer for the appraisal of our manuscript and their comments.

  1. The study demonstrates molecular changes and receptor selectivity. However, predicting the absence of side effects based solely on these findings is challenging. Functional testing in relevant biological systems would be important to confirm whether these compounds avoid predicted side effects, such as bradycardia, hypotension, or sedation. The authors could consider testing these compounds in vascular smooth muscle cells or neuronal models to address this. If testing these models is not feasible, acknowledging this limitation in the discussion would be valuable.

We have previously addressed the role of Gob signalling in the CNS and its centrality to pain relief without undesired effects on heart rate, blood pressure or respiration in vivo (Wall et al. 2022). In the revised manuscript, we have added a phrase to the Introduction to highlight this aspect (lines 71-73). It now reads, “BnOCPA’s signalling dissection has demonstrated that its ability to inhibit excitatory synaptic transmission without causing neuronal membrane hyperpolarisation is due to the propensity to couple to Gob(Wall et al. 2022). Gob signalling activated by BnOCPA was validated as paramount for avoiding depression of heart rate, blood pressure or respiration in urethane-anaesthetised, spontaneously breathing adult rats”.  

  1. Again, functional experiments testing the outcomes of MIPS521 and BnOCPA in pain-related cellular models would significantly strengthen the manuscript. For example, assays measuring neuronal firing rates or calcium signaling in DRG neurons under pain-like conditions could demonstrate their analgesic potential: authors could test the effect of combination of MIPS521 and BnOCPA on relieving simulated pain states. The results would provide critical evidence to support their proposed therapeutic effects

We thank the reviewer for the suggestion. We agree that the experiments we previously reported for BnOCPA should be repeated in the presence of MIPS521 to strengthen our findings. This will be the topic of future work.

  1. The authors state that the interaction between MIPS521 and BnOCPA may reduce the efficacy of BnOCPA in activating Gob proteins. This trade off could be a limitation and the manuscript would benefit from further discussion on how this affects the therapeutic potential of these compounds.

In the Discussion, we have added a piece of text about probe dependency to better frame our results in the context of allosterism (lines 366-371). A further comment has been added to reflect our view that the MIPS521 effect on BnOCPA’s signalling would not represent an obstacle to painkiller development but represents a parallel approach, based on adenosine tone, for analgesia (lines 371-375).

  1. Figures 2e and 3a do not have dot plots or statistical significance annotations. Also, the number of replicates and experimental repetitions is not clearly indicated for all figures.

We have amended Figures 2 and 3 to show standard deviations.

  1. The methods section includes a substantial amount of technical details, but some descriptions are difficult to follow. Reorganizing or rephrasing parts of this section could improve clarity and accessibility for the reader.

We acknowledge that Method sections can be inherently technical. We have now explained the reported acronyms and added titles to specify the discipline involved (i.e. pharmacology, chemical synthesis and computational modelling). We hope this can help the accessibility for the reader.

Round 2

Reviewer 1 Report

Comments and Suggestions for Authors

The authors have provided comprehensive and satisfactory responses to all the major concerns raised during the previous round of review, and I believe the article can be published in its current form.